

# Knowledge graph augmentation: consistency, immutability, reliability, and context

Savaş Takan

Artificial Intelligence and Data Engineering, Ankara University, Ankara, Türkiye

## ABSTRACT

A knowledge graph is convenient for storing knowledge in artificial intelligence applications. On the other hand, it has some shortcomings that need to be improved. These shortcomings can be summarised as the inability to automatically update all the knowledge affecting a piece of knowledge when it changes, ambiguity, inability to sort the knowledge, inability to keep some knowledge immutable, and inability to make a quick comparison between knowledge. In our work, reliability, consistency, immutability, and context mechanisms are integrated into the knowledge graph to solve these deficiencies and improve the knowledge graph's performance. Hash technology is used in the design of these mechanisms. In addition, the mechanisms we have developed are kept separate from the knowledge graph to ensure that the functionality of the knowledge graph is not impaired. The mechanisms we developed within the scope of the study were tested by comparing them with the traditional knowledge graph. It was shown graphically and with t-test methods that our proposed structures have higher performance in terms of update and comparison. It is expected that the mechanisms we have developed will contribute to improving the performance of artificial intelligence software using knowledge graphs.

## INTRODUCTION

Since time immemorial, acquiring, storing, and managing knowledge has been one of the main goals of humanity. Today, thanks to developing technologies, information is multiplying very rapidly. Therefore, it becomes difficult to process, infer and use information. Most of these problems are related to how knowledge is represented. One of the most widely used knowledge representation methods is the knowledge graph (KG).

KGs have emerged as an essential area in artificial intelligence in the last decade (*Rajabi & Etminani, 2022*). A KG can be a directed, labeled, multi-relational graph with some form of semantics (*Kejriwal, 2022*). A KG, or a semantic network, is a graphical representation of real-world entities and relationships, objects, events, situations, or concepts and their relationships. A KG is essential for storing and making inferences from it.

In recent years, KGs have been widely applied in various domains. In parallel, there have been studies on their integration with various domains. These include the creation of semantic KGs for news production, distribution, and consumption in digital news platforms (*Opdahl et al., 2022*), the integration of heterogeneous knowledge sources in the

Corresponding author
Savaş Takan, savastakan@gmail.com

creation of large KGs and artificial intelligence (AI) systems to be more explainable and interpretable (*Rajabi & Etminani, 2022*), the application of KGs in manufacturing and production, reasoning technologies in KGs (*Chen et al., 2020*), the Semantic Web (*Ryen, Soylu & Roman, 2022*), applying machine learning, rule-based learning and natural language processing tools and approaches (*Verma et al., 2022*), and how statistical models can be trained on large KGs and used to predict new facts about the world (*Nickel et al., 2016*).

Although the KG is a very convenient tool for storing knowledge in artificial intelligence, it has some essential requirements and shortcomings, no matter which field it is used in. These shortcomings can be summarized as the problem of automatically updating all the information that affects a piece of knowledge when it changes, the inability to sort information, the inability to keep some information immutable, and the inability to make a quick comparison between information (*Kejriwal, 2022*; *Troussas & Krouska, 2022*; *Noy et al., 2019*). In our work, reliability, consistency, immutability, and context mechanisms are integrated into the KG to contribute to solving these problems. However, it should be emphasized that these mechanisms are not extensions (*Choi & Ko, 2023*; *Simov, Popov & Osenova, 2016*) because the purpose of integrating these mechanisms into the KG is to improve the performance (*Macdonald & Barbosa, 2020*; *Yang et al., 2022b*; *Cannaviccio et al., 2018*) of existing KGs by contributing to the solution of their basic problems. In this respect, referring to this integration as KG augmentation is considered more appropriate.

The KG must be always consistent (*Mu, 2015*). This consistency may be lost if any information changes. To restore coherence, all the information connected to the changed information must change. This is because a change in the elements that support a piece of knowledge, with a chain effect, calls into question the reality of all the elements supported by that knowledge. Time is vital to ensure consistency in the knowledge change (*Terenziani, 2000*). In addition, consistently keeping knowledge helps to reduce complexity (*Liberatore & Schaerf, 2001*). The classical knowledge structure can find changing knowledge by cause-effect and inference. However, since such methods do not have stamping and tracking, they are complex and can lead to overlooking information that needs to change. Moreover, if this inference is global, it will have performance problems, and if it is local, it will return conflicting information because it cannot capture change. At the same time, there are severe performance penalties when erroneous information is removed, new information is added, or existing information is modified.

Another requirement for the KG is to ensure the ordering of knowledge (*Porebski, 2022*). We have integrated a reliability mechanism into the KG to fulfill this requirement. Accordingly, the more reliable elements supporting a piece of knowledge, the more reliable that knowledge is considered to be. In the opposite case, the knowledge in question is interpreted as doubtful. Thus, ranking between knowledge becomes possible.

Another requirement in the KG is the comparison of two pieces of knowledge (*Wu et al., 2021*; *Jabla et al., 2022*). It is very important that this comparison can be made very quickly. In our work, we integrate a hashing mechanism called context into the KG, which allows us to determine the identity of two pieces of knowledge in O(1) time. Context allows

the disambiguation of a piece of knowledge by looking at its contexts. For example, Jaguar refers to both an animal and a programming language. The ambiguity about which of these is expressed in a KG can be resolved by comparing its constituent knowledge, thanks to the context augmentation we have developed.

Another vital element of the KG is that knowledge can be immutable (*Cano-Benito, Cimmino & García-Castro, 2021*; *Besançon et al., 2022*). For example, while the people who buy or read a book can change, the book's title and the author must be immutable. In other words, some elements can change knowledge, and others cannot.

The proposed augmentation ensures consistency by marking the knowledge as soon as it changes and updating the associated knowledge to run in the background at any time. To ensure that the knowledge is immutable, a structure has been created to store immutable and mutable data. Regarding the reliability of the knowledge, an information hierarchy has been developed in the system. Regarding context, the summarization function provides unique hash values for existing contexts. Thus, when there is a match between different contexts of two pieces of knowledge, it can be quickly recognized that they have the same context.

In the study, the research on the subject is given respect, and then the methodology of the proposed plugins is explained. Then, the plugins are explained in detail, and their advantages and disadvantages are presented.

## RELATED WORK

There is a vast literature on the KG. There are primarily many review articles on the topic (*Chen et al., 2021*; *Cambria et al., 2021*; *Chen, Jia & Xiang, 2020*; *Issa et al., 2021*; *Dai et al., 2020*).

Knowledge graph augmentation adds missing facts to an incomplete knowledge graph to improve its effectiveness in web search and question-answering applications. State-of-the-art methods rely on information extraction from running text, leaving rich sources of facts such as tables behind. Focusing on closing this gap in their work (*Macdonald & Barbosa, 2020*), the researchers developed a neural method that uses contextual information surrounding the table in a Wikipedia article. In a different work (*Yang et al., 2022b*), a general knowledge graph contrastive learning framework (KGCL) and a knowledge graph augmentation scheme that mitigates knowledge noise for knowledge graph-enhanced recommender systems are proposed.

In a recent work on the topic, a data-efficient method for multilingual named entity (MNE) resources with more languages was developed (*Severini et al., 2022*). A different study developed a supervised approach to extract missing categorical features in Web markup (*Tempelmeier, Demidova & Dietze, 2018*). In another article, a new model is proposed that effectively links new entities and existing KGs through a pre-trained language model using two learning methods (*Choi & Ko, 2023*). *Sagi, Wolf & Hose (2019)* investigated the prevalence of novel entities in news feeds to determine how much information is novel and not grounded. In another study, a strategy for enriching WSD knowledge bases with data-driven relations from a gold standard *corpus* was presented,

and it was shown that the accuracy in the WSD task increased statistically significantly (*Simov, Popov & Osenova, 2016*).

General solutions to augment KGs with facts extracted from Web tables aim to associate pairs of column columns with a KG relation based on the matches between pairs of entities in the table and facts in the KG. Motivated by the shortcomings of these approaches, researchers in one study (*Cannaviccio et al., 2018*) proposed an alternative solution that exploits patterns emerging in the schemas of a large *corpus* of Wikipedia tables. In another study (*Nguyen et al., 2023*) introducing SocioPedia+, a real-time visual analysis system for social event discovery in time and space domains, a social knowledge graph dimension was added to the multivariate analysis of the system, making the process significantly improvable.

On the other hand, many studies focus on consistency in KG. In one of the most influential early studies, two new complementary features on constraints in a network were developed (*van Beek & Dechter, 1997*). The authors suggest that these features can be used to decide whether it would be helpful to pre-process the network before a callback search. In a different study, tools for consistency checking were found to provide an opportunity to reduce minor inconsistencies in the Gene Ontology (GO), and redundancies in its representation (*Yeh et al., 2003*). Another study presented a general, consistency-based framework for expressing belief change (*Delgrande & Schaub, 2003*). With this framework, other belief change operations, such as updating and deleting, can also be expressed. In another study, a measurement parameter was developed to quantify the amount of inconsistency in probabilistic knowledge bases (*Muiño, 2011*). The study measured inconsistency by considering the minimum adjustments in the degrees of certainty of statements (*i.e.*, probabilities in this article) necessary to make the knowledge base consistent. In a different study, *Mu (2015)* proposed a measure for the degree of responsibility of each formula in a knowledge base for the inconsistency of that base. This measure is given in terms of the minimum, inconsistent subsets of a knowledge base.

A different study on the topic includes studies that address the central problem of the computational complexity of consistency checking (*Grant, Molinaro & Parisi, 2018*), as well as a graph-based approach to measuring inconsistency for a knowledge base (*Mu, 2018*) better to understand the nature of inconsistency in a knowledge base. Another recent study starts from the challenges of the belief revision process (*Bello López & De Ita Luna, 2021*). Accordingly, one of the most critical problems is how to represent the knowledge base K to be considered and how to add new information. In this article, an algorithmic proposal is developed to determine when (K E (K *)) is inconsistent.

Besides consistency, context is central to many modern safety and security-critical applications. In a different study, the phrase similarity of human comments was determined using four different methods, including item matching, linguistic collocation approaches, and wordnet semantic network distance (*Stock & Yousaf, 2018*). The method that incorporates context is said to be the most successful of the four methods tested, selecting the same geometric configuration as human respondents in 69% of cases. In another study on the context in KGs, a formal approach to achieve contextual reasoning

was developed based on the lack of formal integration of knowledge and context in existing context-aware systems (*Alsaig, Alagar & Nematollaah, 2020*).

In the literature, there are many studies on the ordering of nodes in graph theory (*Sciriha & da Fonseca, 2012*; *Nirmala & Nadarajan, 2022*; *Huang et al., 2021*; *Christoforou et al., 2021*). However, as far as we know, there needs to be research on ordering in the KG. At the same time, although the issue of immutability in data structures has been frequently studied (*Chowdhury et al., 2018*; *Ozdayi, Kantarcioglu & Malin, 2020*; *Stančić & Bralić, 2021*; *Balakrishnan, Ziarek & Kennedy, 2019*), there is no research on immutability in KGs. In addition, although several studies focused on reliability and ranking in the KG (*Seo, Oh & Lee, 2020*; *Yang et al., 2022a*; *Jiang et al., 2022*), these studies are not directly related to the topic of our article. Similarly, only some studies focused on hashing in the KG (*Khan et al., 2023*; *Wang, Shang & Qiao, 2020*). Still, the existing studies in the literature are separate from the plugins we developed in our article.

As can be seen, studies on KG in the literature have covered a wide range of topics. Studies have generally focused on integrating the KG into other domains. Studies focusing on consistency in the KG have generally developed complex solutions in the literature. In the limited number of context-oriented studies in the literature, application-based solutions have been developed without any change in the structure of the KG. Our study differs from the existing studies in the literature that focus on consistency and context in the KG by providing these extensions with hashing technology. This is because no studies in the existing literature integrate invariance, consistency, reliability, and context into the KG using hashing technology. The main contribution of our work to the literature is to show that four different properties can be integrated into the KG with a simple mechanism (Hashing). In this respect, our work is expected to contribute to the literature on better representation of knowledge, the solutions created, and the development of artificial intelligence software using KGs.

## MATERIAL AND METHODS

In our work, consistency, context, reliability, and immutability mechanisms are integrated into the KG to perform the operations of where existing knowledge comes from, by whom it is supported, the rate of support, ranking, and whether it is a modifiable or immutable and automatic update. Unlike the literature, these augmentations were developed using the hashing mechanism. This is because Hashing technology, a straightforward mechanism, offers the possibility to provide four different properties quickly. In our work, a "Knowledge" model provides consistency, immutability, reliability, and context augmentations to the KG.

Thanks to the hashing mechanism, it is possible to check whether the relationships and data in the information have changed. Relationships that are checked whether they change are called constant relationships, and relationships that are not checked are called variable relationships. On the other hand, data is constantly checked and therefore considered constant in the KG. Figure 1 shows the general features of the KG we developed.

The hash mechanism provides immutability control in the KG. Here, a hash is a hash of immutable relations and data. The hash is calculated and added to the hash set at any time.

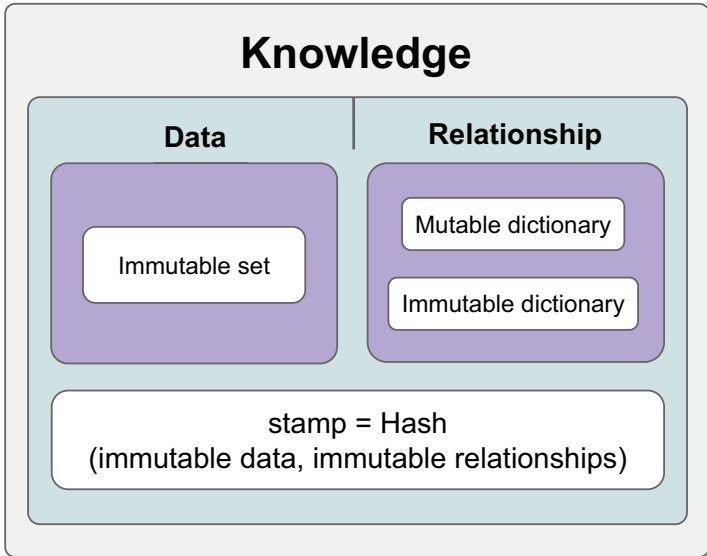

**Figure 1  The general scope of knowledge.**

In the information structure we have developed, this is done with the lock() function. Then, a new hash value is calculated and compared with the old hash values in the hash set to check for any changes in the information. After that, if there is a change in the relations or data of the information, a different hash value will be output, so it can be automatically determined whether the information has changed and, if so, its position. If the results are equal, the structure has not been changed; if the results are not equal, it means the structure has been changed. This is done with the isLock() function. The general structure of the lock and islock state of the information is shown in Fig. 2.

An example hash-finding formula is as follows. Here, the value i indicates how many of the n fixed data are expressed. The value j indicates how many of the m fixed relations are expressed.

Calculating the information hash value:

$$hash\left(\sum_{i=0}^{n} Immutable\ data_i + \sum_{j=0}^{m} Immutable\ relationship_j\right) \qquad (1)$$

when a piece of information is deleted, modified, created, or added, the information that depends on it is recalculated and updated by Algorithm 1 to maintain consistency. In parallel, Algorithm 1 can detect if there has been a change in the KG. Algorithm 1 ensures changes are propagated to all low-complexity points in the KG. In addition, the same algorithm can also be used to find where the changes in the KG have occurred. In the algorithm, updates are performed on the invariant relations in the KG. In other words, variable relations are not taken into account. This algorithm was developed using depth-first search, dynamic programming, and topological ranking.

Since the KG is a cyclic graph with multiple transitions, nodes, and edges are swapped to traverse all transitions. Thus, all edges can be traversed. In this way, the whole system is traversed with O(E) complexity. As a result, the whole system can be updated with linear

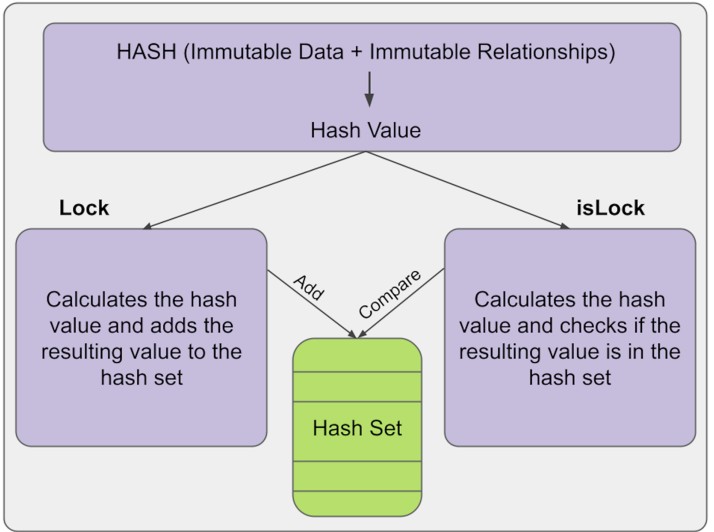

**Figure 2 Lock and unlock states.**

---

**Algorithm 1** The algorithm below updates all affected nodes, and edges, whenever there is a change in any node.

stack ← [*startEdge*]

*visited* ← []

**while** *stack* **do**

    **for all** neighbor_edge ∈ graph.edges(edge[1]) **do**

        **if** neighbor_edge ∉ visited **then**

            visited.append(neighbor_edge)

            **if** edge ∉ parent[neighbor_edge] **then**

                parent[neighbor_edge].append(edge)

                cost ← weight_cost[edge] + graph[neighbor_edge]['weight']

                weight_cost[neighbor_edge] + = cost

            **end if**

        **end if**

    **end for**

**end while**

---

complexity. In the KG, the update can be determined according to the depth parameter given by the user. Thus, the user can determine how many depth units can be updated.

## LIMITATIONS

To physically test the model we developed, four STM32 and Lora Modules were used, and tests for readability, storage, and data manipulation were performed. As a result, it was determined that the system could physically operate without problems. However, for

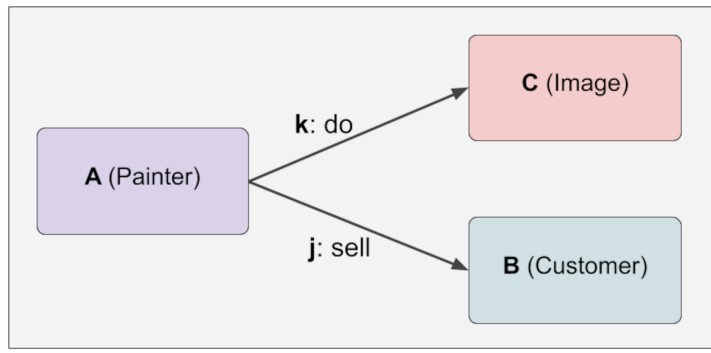

**Figure 3** **Immutability.**               

financial reasons, more comprehensive and holistic tests could not be carried out at this stage, and it was impossible to test the model we developed on large systems. However, such a test in our work is considered necessary and valuable. For this reason, more comprehensive applications will be realized by providing the necessary resources.

Immutability, reliability, consistency, and contextualization are not elements that can be easily tested. For this reason, in our study, we have tried to prove the applicability of these elements through example scenarios. In future studies, running it on real scenarios would be helpful.

# KNOWLEDGE GRAPH AUGMENTATIONS

This section describes each plugin we developed for the KG and proves their functionality by testing them with example scenarios. Thus, it is shown that our KG augmentations can be used in various software processes.

## Immutability

To illustrate the uncontrolled relationship, five different pieces of information are constructed below. A has an outward relationship with C and B through k and j. B, C, j, k have no relationship at this stage. These five pieces of knowledge are generated in Fig. 3:

The lock function has yet to be executed in the phase shown above. Therefore, no immutability mechanism has been activated, meaning the hash values will be shown as Null. In the JSON representation above, there are four values. The first is the checked data. The second is the checked relations, the third is the unchecked relations, and the fourth is the hash value. By calling the lock function of A with the following command, the system is locked and thus made unalterable. The command to call the lock function of A is as follows:

A.lock()

After the Lock function is applied, the JSON format view of the structure follows. The point to note here is that hash values are entered. Since C and k information is dependent on A, when A information is made immutable, this information also becomes immutable. On the other hand, since j and B are not checked (they have a variable relationship), they are not fixed, and the hash value remains Null. This can be easily seen in Formula (1).

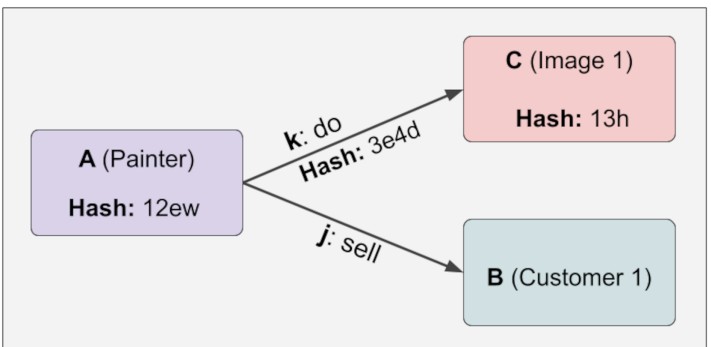

**Figure 4** The calculation in Formula (1) and Algorithm 1.

Furthermore, with the calculation in Formula (1) and Algorithm 1, A can determine whether the information in k and C has changed, and if so, which information has changed in Fig. 4.

At any time, a new piece of information can be added linked to the variable relation, and the hash value will not change even if the lock function is executed. This provides design flexibility. Because some relations are fixed while others are variable. For example, the painter of a work of art is fixed, while the customers who buy this work of art are variable. It is challenging to create this structure in the KG. Below, it is shown in Fig. 5 that hash values do not change even if the relations we do not control (variable) change:

Suppose new information is added to the immutable relation on demand. In that case, the hash values are reconstructed, and when these new values are compared with the old ones, it will be seen that the newly created hash values are different from the old ones. The point we want to draw attention to here is that the hash value of the A information will change when a new D information is added to the above example. This is shown below in Fig. 6:

Diversity for uncontrolled relations needs to be present in the KG. This significantly affects the design manipulations. For example, if j and B did not have varying relationships, customer 1 information would remain in the system. Also, the hash values of the entire system would have to be recalculated in case of any changes. Furthermore, since there are no lock and isLock functions in the KG, the system can only be fixed manually or created from scratch. This can lead to serious time and space losses.

## Reliability

The reliability augmentation consists of the sum of invariant relations in the KG. In this way, the reliability mechanism allows information to be ranked. Information with a high trust value is more secure and ranks higher.

In reliability augmentation, the reliability of a piece of information is related to the number of immutable relations it has. That is, the more immutable relations a piece of information has with other information, the more reliable it is. It is called suspect information if a piece of information has no fixed relationships. The following example

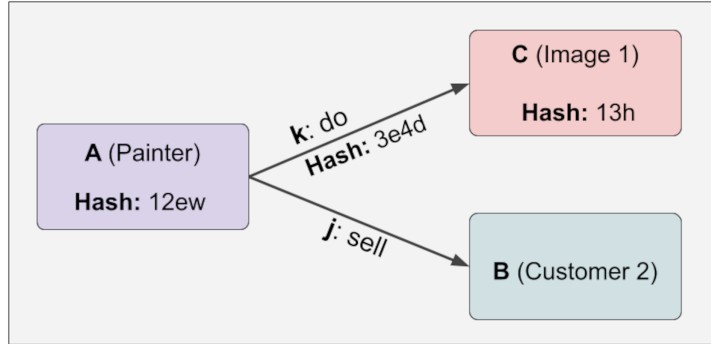

**Figure 5 Hash values do not change even if the relations we do not control (variable) change.**

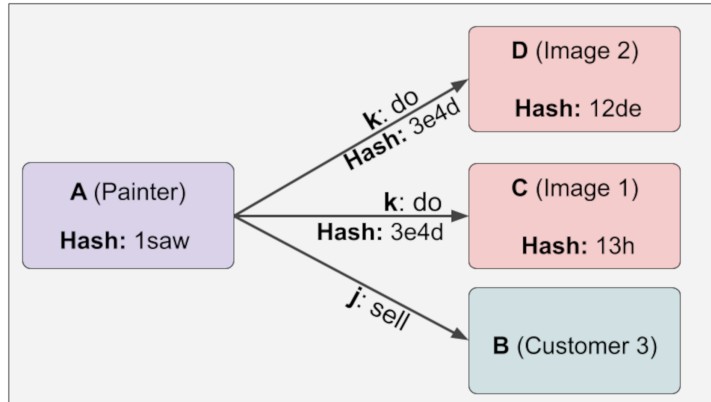

**Figure 6 The hash value of the A information will change when a new D information is added.**

shows a JSON representation of C, which has no fixed relationships. Here the reliability value of C is 0.

**C:** {{'Image 1'}, Null, Null, {13h}}

The representation of Z information with more than one constant relationship is shown in Fig. 7. Here, the reliability value of Z is 2. Regarding reliability, if the user enters a depth parameter, the calculations are made up to that depth. For example, if the depth parameter of Z is 1, the reliability value will also be 1. This feature has been developed to reduce time and space complexity significantly.

The KG has no practical and simple reliability mechanism in the sense we have developed. It is, therefore, not possible to rank trustworthiness. This prevents a trust-based ranking mechanism. On the other hand, the trustworthiness mechanism we have developed can be applied practically and simply to the KG, thus efficiently addressing the need for trust-based ranking when necessary.

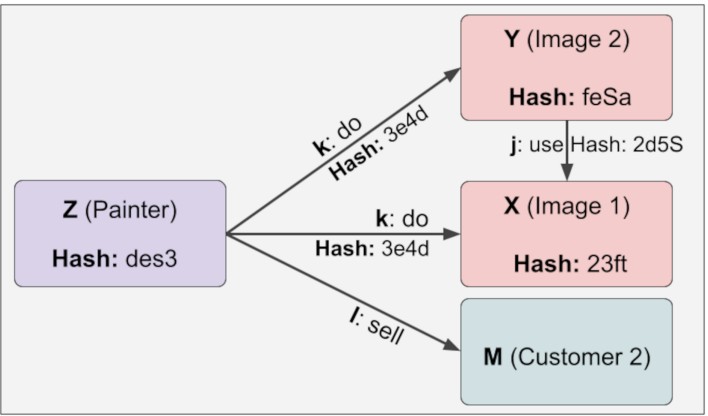

**Figure 7 The representation of Z information with more than one constant relationship.**

## Consistency

This section explains the consistency augmentation in the KG through an example scenario. First, five pieces of information are created. At the time of creation, they have no fixed or variable relationships. Below, the creation of the information is shown in JSON format.

**K1:** {'A 36-year-old man stabbed his ex-fiancée to death.', Null, Null, Null, Null}

**K2:** {'23 years in prison sentence requested for a man who stabbed his ex-fiancée to death.', Null, Null, Null}

**K3:** {'Man who stabbed his ex-fiancée to death is released in good condition after the first hearing.', Null, Null, Null}

**K4:** {'Women's rights activists protested this decision in front of the court.', Null, Null, Null}

**K5:** {'Feminism is spreading.', Null, Null, Null, Null}

Once information is created, a cause-and-effect relationship is established between them. If a piece of information has no relationship, it is not reliable. For example, in the commands below, the cause of the fifth information is the relationship between the fourth, the cause of the fourth is the relationship between the third, the cause of the third is the relationship between the second, and the cause of the second is the relationship between the first. The disappearance of the fourth piece of information would remove the reliability of the fifth piece of information and make it suspect. Below, after the cause and effect relationships of the information have been entered, the relationships between the information are shown in JSON format, locked with the lock function.

**A1:** {'why', Null, Null, {12fK}}

**K1:** {'A 36-year-old man stabbed his ex-fiancée to death.', Null, Null, {76Tf}}

**K2:** {'A man who stabbed his ex-fiancée to death has been sentenced to 23 years in prison.', {A1: K1}, Null {23wS}}

**K3:** {'The man who stabbed his ex-fiancée to death was released in good condition at the first hearing.', {A1: K2, **A1:** K1}, Null, {23dS}}

**K4:** {'Women's rights activists protested this decision in front of the court.', {A1: K3}, Null, {P3se}}

**K5:** {'Feminism is spreading.', {A1: K4}, Null, {wqq2}}

When an error or change occurs in the information itself or in any of the fixed information that supports it, the model finds the source of the change, removes that source from the context, and updates all the information associated with that source depending on the depth parameter. This ensures consistency in the system.

The consistency concept in the plugin we developed focuses on changes in the KG copy and in the KG itself. A change in any information in the KG will cause every piece of information in the KG to be updated and make it possible to update changes in its copies on demand.

## Context

Below, four relationships are created to explain the context of a piece of information.

**A1:** {{'why'}, Null, Null, Null, Null}

**K1:** {{'data1'}, Null, Null, Null, Null}

**K2:** {{'data2'}, A1:K1, Null, Null}

**K3:** {{'data3'}, {A1:K1, A1:K2}, Null, Null}

**K4:** {{'data4'}, {A1:K3}, Null, Null}

I have shared the context of the above knowledge below for you to review. Here, Knowledge1 has no context but persists in the model. This means that there is no invariant relationship to verify Knowledge1. Whether or not any knowledge that has no invariant relationship and is therefore not ordinarily reliable is considered reliable is at the discretion of the creator of the KG.

As seen in Fig. 8, the context of Knowledge4 includes Knowledge1, Knowledge2, and Knowledge3. Since the hash value will be specific to the graph when calculating the context, the context of Knowledge4 in the figure above will be specific to Knowledge1, Knowledge2, and Knowledge3 and the relationship between them. As can be seen in Fig. 3, the values held in the hash set also determine the context. In other words, since a piece of knowledge can have multiple contexts, it is possible to create the contexts of that knowledge by assigning the desired summaries to the hash set. By looking at these hash values, it can then determine whether one piece of information is compatible with the context of another. This removes many ambiguities about information.

The plugin we developed supports the context mechanism for comparing information in the KG. This makes it possible to compare information in the KG easily. As in real life, the value of a piece of information can vary according to many different contexts. This can be easily realized in the plugin we have developed.

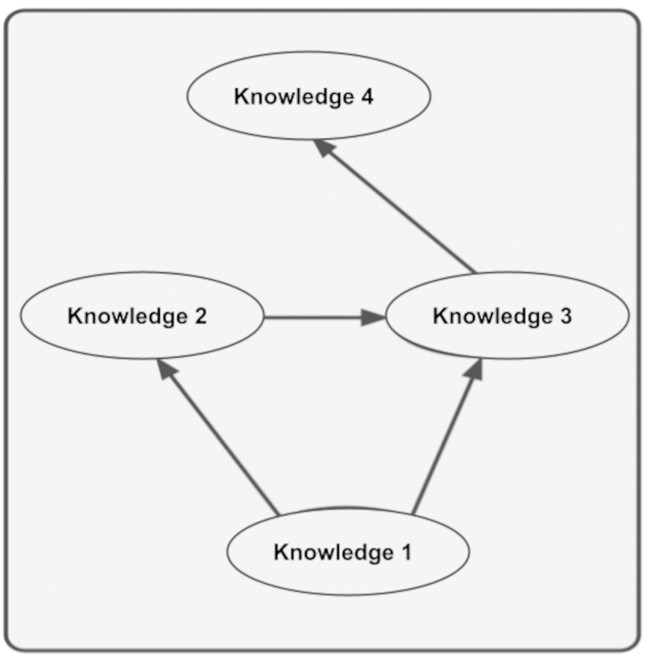

**Figure 8 Knowledge's context structure.**

## EXPERIMENT

To test the augmentations we developed, we created the experimental setup in Fig. 9. In this experiment, persons A, B, C, D, and E are created, and the relationships between these persons are shown. In the relationships between people, the red arrows cannot be changed, and the blue arrows can be changed. For example, being an artist or a father is a fixed relationship. In contrast, being a moviegoer, the city one lives in, one's friends, or hobbies are relationships that can change depending on one's choice. The more fixed relationships person A has, the more trustworthy he/she is considered to be. For example, in Fig. 9, A's credibility is 3, and B's is 4. In this case, B is considered more trustworthy than A. Whenever an update is made to B, the constant relations between the labels "female, E and Ankara University" that support B are also updated. This mechanism is not present in the graph data structure.

For example, if we want to change the cycling hobby, we need to update people D and A who are affected by that hobby. Normally we have to do this manually, which poses a problem for the consistency of the KG. For example, we need to remember to add the information to the KG or more time to add the information, which can lead to various problems. This can lead to various inference problems in the KG. Therefore, an algorithm has been developed that automatically updates any change on the fly. Thus, in the experimental example, A and D were updated automatically.

Finally, when we want to compare any two people, for example, C and E are people with the same occupation and living in the same city. C and E are considered the same if this information is recorded as context. But C is A's father, and E is B's sister. From this point of view, C and E are treated as different people because they have different contexts.
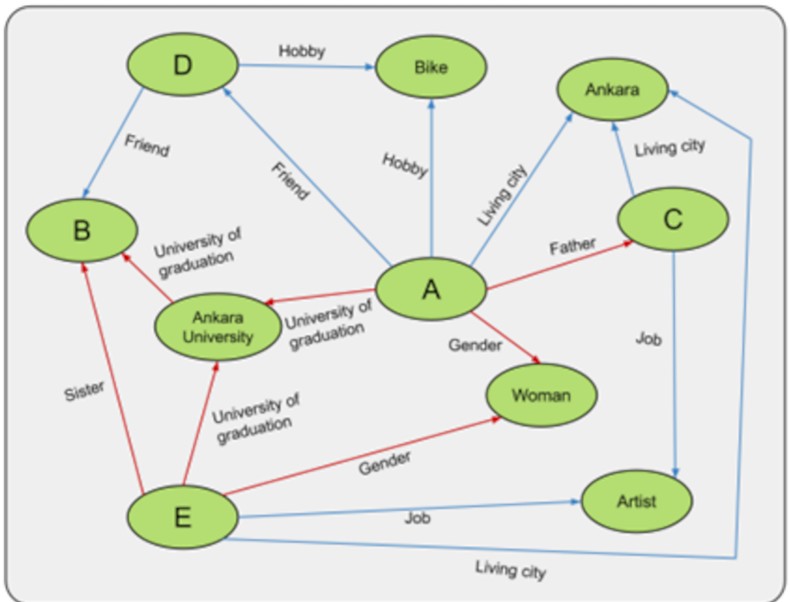

**Figure 9 General representation of the experiment.**

Thanks to the hashing mechanism used in our experiments, it is possible to determine in a very short time which contexts they are in and which they have in common.

Apart from the above scenario, on a computer with Intel i7 16 GB ram, random KGs from 1 to 3,000 knowledge were generated using Python in the Networkx library. On top of that, the method we developed in this study was tested by comparing it with the traditional method in terms of update and search mechanisms.

In the experiment, we first focus on the update rate of the knowledge graph. The reason for testing the update rate is that it has a strong relationship with consistency, immutability and reliability. Reliability ensures that information is linked together, and immutability ensures that information is updated quickly whenever there is a change in the information. The totality of this fast updating process is consistency. From this point of view, our experiment demonstrated the time it takes to maintain consistency in the knowledge graph after a change occurs.

Looking at the experiment results, linear time is required to access information. After accessing the information in question, Deep First Search or Breath First Search must be used to update the values. Since this also takes linear time, a total of $O(n^2)$ time is needed. On the other hand, the algorithm we developed uses only Deep First Search because it is updated instantly, and therefore its complexity is in linear time. The experiment results are shown in Fig. 10. As a result of our experiments, the *P* value of the t-test was 1.30e−06. Herefore, there is a significant difference between the two values.

The graph below looks at the context in the knowledge graph. The main feature of context is the comparison of the similarity of the relationships of two different nodes.

Finding two pieces of information first and then looking at the properties of the found information leads to exponential complexity. On the other hand, since our algorithm uses the hashing mechanism, comparing two pieces of information takes place in constant time

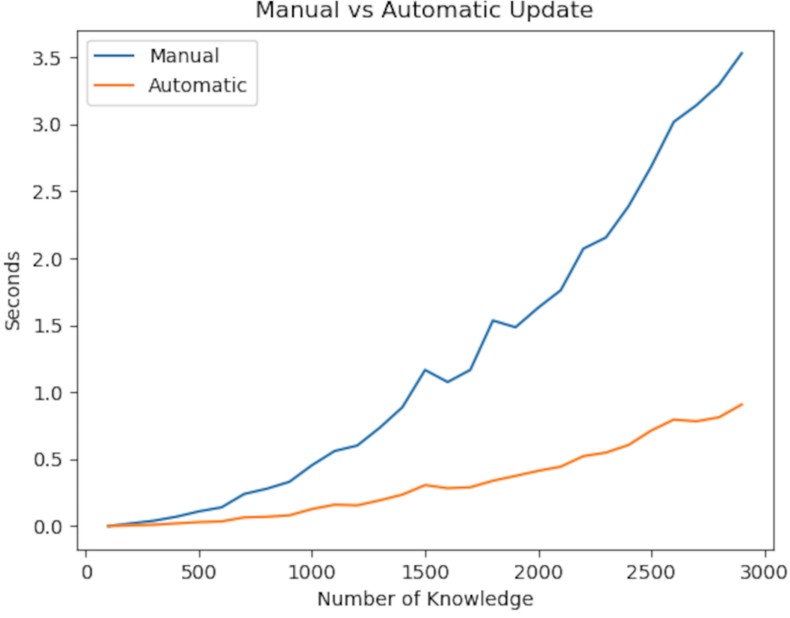

**Figure 10 Updating speed of the knowledge graph with improved augmentations.**

in Fig. 11. This fixed time is the length of the context set. As a result of our experiments, the
*P* value of the t-test was 6.39e−05. Therefore, there is a significant difference between the
two values.

Table 1 compares with the knowledge graph to illustrate the advantages of the
developed augmentations. Based on our experiment, we can say that our augmentations
provide time benefits by eliminating some important shortcomings in artificial
intelligence.

## EVALUATION

This section presents the advantages and disadvantages of the augmentations we
developed for the KG.

The advantage of the immutability augmentation is that the information in the KG is
stamped as changed/unchanged, making it easy to identify which information has
changed. While there is a wide range of work on immutability in the data structure, there
must be work on immutability in KGs.

The immutability plugin contributes to keeping the KG consistent by allowing
information to be easily updated. This contribution is referred to as consistency
augmentation in this article. Thanks to Algorithm 1, the consistency augmentation hovers
over all changed information and ensures that this information is updated quickly. This
function is executed automatically when a piece of information changes and updates all the
information it affects based on that change. There is a wide variety of work on consistency
in the KG. However, these studies have yet to use a hashing mechanism. At the same time,
almost all of the studies in the literature involve very complex procedures.

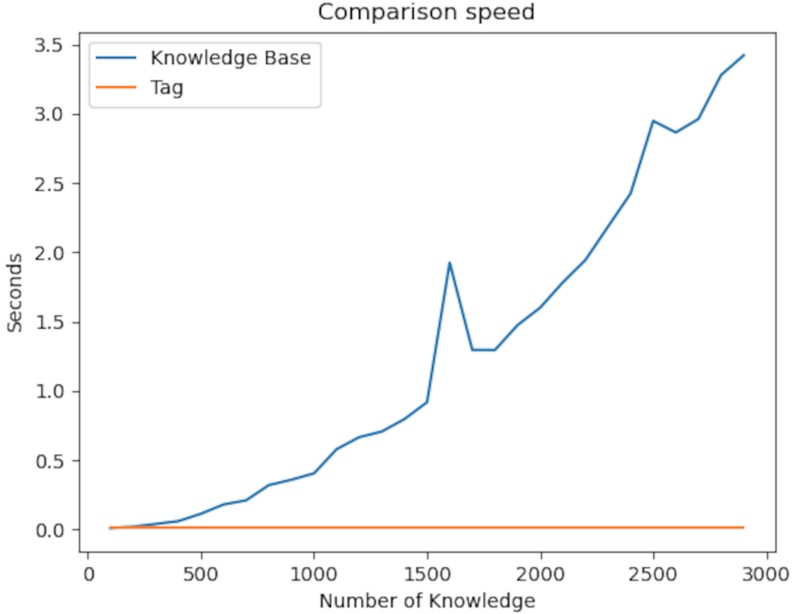

**Figure 11 Comparison speed of the knowledge graph with the plugins developed.**

**Table 1 Comparison of knowledge graph and tag mechanism.**

| Type | Update | Immutability | Sorting | Comparison |
|---|---|---|---|---|
| Knowledge graph | Manual | No | $O(n^2)$ | $O(n^2)$ |
| Tag | Automatic | Yes | $O(1)$ | $O(1)$ |

Credibility augmentation allows for the ranking of information. Information ranking reveals the importance of two or more pieces of information. Ranking information in the KG according to its importance provides the advantage and flexibility to compare it. Few studies on trustworthiness in KGs have used hashing mechanisms in the literature. At the same time, almost all existing studies involve rather complex procedures.

Context augmentation allows a comparison between two pieces of information. Context augmentation allows us to determine whether the information is the same by looking at the hash values. Thanks to the hash set, the information has more than one context, and again thanks to the hash value, the context in which the information is located can be determined. This gives the KG the advantage of flexibility and abstraction. Moreover, the time complexity is $O(1)$ due to the comparison with the hash algorithm. Although there are several studies on the context of KGs, they have yet to use the hashing mechanism. At the same time, almost all existing works involve very complex procedures.

In our work, the disadvantage of the four augmentations developed for the KG is that the hash values of all the information the KG is linked to are stored due to the hashing mechanism. Here, a hash value of length N × (256 bytes) is stored if the information has N links. This slightly increases the space complexity. Another aspect is the runtime of the

update function, which is O(E) complexity. The update can be increased or decreased with the diameter parameter. This has a significant impact on the complexity. Considering the contributions of the augmentations we have developed to the KG, these two issues, which can be expressed as disadvantages, can be ignored.

## CONCLUSION

In our work, consistency, context, reliability, and immutability mechanisms are integrated into the KG modularly to perform the operations of where existing knowledge comes from, by whom it is supported, the rate of support, ranking, modifiability or immutability, and automatic update. The hashing mechanism was used in the development of these plugins. This is because hashing technology, a straightforward mechanism, can provide four different properties quickly. In our work, a "Knowledge" model provides consistency, immutability, reliability, and context to the KG.

The first of our proposed extensions, immutability, ensures that all associated information is immutable when one piece of information is immutable. This guarantees information reliability. The hash information changes whenever there is a change, so it is immediately possible to identify where the change occurred. The level of trustworthiness is related to the amount of trustworthy information that supports the information. This allows information to be ranked according to its trustworthiness. Consistency refers to the fact that whenever there is a change in the KG, all affected information is immediately updated. The context consists of all the information about a piece of knowledge and its relationships. The different contexts are calculated and stored in a context array, and the information can be checked for relevance to other contexts by looking at the context array.

With the augmentations we have developed, additional features have been added to the KG, enabling it to reflect knowledge more comprehensively. The augmentations are expected to contribute to developing artificial intelligence software that utilizes the KG. In a broader sense, our work is expected to contribute to developing the software needed in knowledge representation and a wide range of fields related to knowledge since knowledge is a structure used in every field. In future work, it is planned to realize comprehensive plot implementations of the plugins developed as a proposal.

### Funding
The authors received no funding for this work.

### Competing Interests
The authors declare that they have no competing interests.

### Author Contributions
- Savaş Takan conceived and designed the experiments, performed the experiments, analyzed the data, performed the computation work, prepared figures and/or tables, authored or reviewed drafts of the article, and approved the final draft.

## Data Availability

The source codes are available in the Supplemental Files.

## Supplemental Information

Supplemental information for this article can be found online at http://dx.doi.org/10.7717/peerj-cs.1542#supplemental-information.

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
