# Peer review of "Knowledge graph augmentation: consistency, immutability, reliability, and context"

_PeerJ Computer Science, doi:10.7717/peerj-cs.1542_

## Round 0.1 · original submission · Major Revisions

Dear Author,
Please clearly mention the research motivation and contributions of this work. Also, improve the English language presentation of this manuscript. Thank you.

Reviewer 1 ·

Basic reporting

Please see the "Additional comments".

Experimental design

Please see the "Additional comments".

Validity of the findings

Please see the "Additional comments".

Additional comments

This paper explores knowledge graph extensions and specifically their consistency, immutability, reliability, and context.
The paper discusses a hot topic of the related literature that the reader of this Journal would like to read.
While this is a very interesting paper, I think it is necessary to address some concerns before publication.
Some improvements should be done for a better comprehensive reading.
I would suggest to the authors that include some discussion about explainability for the results. Also, the following issues should be improved:
1. In the abstract, the novelty of this research should be discussed.
2. In the introduction, the motivation and contribution of this paper should be given.
3. The introduction should be based on references.
4. Since the literature review is quite poor and to support several assertions, the authors are advised to use the following references:
-a. Kejriwal M. Knowledge Graphs: A Practical Review of the Research Landscape. Information. 2022; 13(4):161. https://doi.org/10.3390/info13040161.
-b. Troussas C, Krouska A. Path-Based Recommender System for Learning Activities Using Knowledge Graphs. Information. 2023; 14(1):9. https://doi.org/10.3390/info14010009.
-c. Natasha Noy Yuqing Gao Anshu Jain Anant Narayanan Alan Patterson Jamie Taylor Communications of the ACM, vol. 62 (8) (2019), pp. 36-43.
5. A schema describing the research methodology could be helpful.
6. More information about the population could be included.
7. The conclusions should lead to new knowledge. Also, limitations of this research are missing at the moment.
8. The academic writing needs work. The authors should correct several mistakes in the use of the English language.
Concluding, the structure of paper is good, but the main contributions of the paper do not add significant value to the existing body of knowledge in the related subject area.
A suggested contribution is to have a discussion section to compare the presented work with the related work in the literature.
Overall, the paper is well organized. However, it lacks critical discussion in contrast with the related work in the literature and does not provide major contributions to the field.

Reviewer 2 ·

Basic reporting

This paper focuses on the research about the extensions of knowledge graph. The author integrated fundamental mechanisms such as reliability, consistency, immutability, and context into the knowledge graph by using the hashing mechanism.
However, this paper is poor written. The studies in this paper lack of clear explanations and the practical significance needs to be further explained, better presented. The method given in this paper it has limited innovations. There is neither theoretical innovation nor technological innovation.
Concretely speaking, there are several problems noticed in this paper, a list of these problems is given below. Please check the following list and make further discussion and revision seriously.

1 Problem: The research motivation of this paper is not clear. As mentioned in Section 1, the author says that “A model was created with a knowledge ranking mechanism to distinguish more reliable knowledge from unreliable knowledge in the study of knowledge”. But why does the uncertainty of knowledge come into being? What is the impact? In this paper, the author does not answer the above questions well.

2 Problem: The related work only introduces the current research progress of knowledge graph and its application in various fields. No relevant research was introduced around the content of the author's research. As the author said, “the existing literature has identified no study on the development of immutability, consistency, reliability, and context plugins to the knowledge graph”, but is there really no relevant research? If so, what is the significance of this study?

3 Problem: In “Related Work”, the author said to introduce the blockchain hash function to study the knowledge graph, unfortunately, the relevant research on the blockchain hash function has not been listed.

4 Problem: In page 3, the pseudo code has no meaning and cannot explain the key content of the study.

5 Problem: In the last paragraph of page 4, the three functions only obtain hash values, lock them and judge whether they are locked. The algorithm is too simple and has no innovation.

6 Problem: As the author said at the end of Section 1: “To accomplish the empowerment above, the blockchain’s hashing mechanism was utilised”, however, the whole sections 2 and 3 do not reflect the role of blockchain.

7 Problem: There are no experiments in the article, just a few simple examples, which cannot well reflect the validity and practical significance of the research results.

Experimental design

1 Problem: There are no experiments in the article, just a few simple examples, which cannot well reflect the validity and practical significance of the research results.

Validity of the findings

1 Problem: The research motivation of this paper is not clear. As mentioned in Section 1, the author says that “A model was created with a knowledge ranking mechanism to distinguish more reliable knowledge from unreliable knowledge in the study of knowledge”. But why does the uncertainty of knowledge come into being? What is the impact? In this paper, the author does not answer the above questions well.

2 Problem: In page 3, the pseudo code has no meaning and cannot explain the key content of the study.

3 Problem: In the last paragraph of page 4, the three functions only obtain hash values, lock them and judge whether they are locked. The algorithm is too simple and has no innovation.

4 Problem: As the author said at the end of Section 1: “To accomplish the empowerment above, the blockchain’s hashing mechanism was utilised”, however, the whole sections 2 and 3 do not reflect the role of blockchain.

Additional comments

1 Problem: The related work only introduces the current research progress of knowledge graph and its application in various fields. No relevant research was introduced around the content of the author's research. As the author said, “the existing literature has identified no study on the development of immutability, consistency, reliability, and context plugins to the knowledge graph”, but is there really no relevant research? If so, what is the significance of this study?

2 Problem: In “Related Work”, the author said to introduce the blockchain hash function to study the knowledge graph, unfortunately, the relevant research on the blockchain hash function has not been listed.

Reviewer 3 ·

Basic reporting

1. The terms of constancy, reliability, consistency, and context in the article should be quite clear. I suggest you elaborate on the terms with sufficient examples.
2. The author used the concept of hashing-mechanism to apply the proposed extension. Please tell why you used the approach?

Experimental design

You described some examples and explanations in the Result section. However, the methods and scenarios used in the experiment are not already clear, especially on how to measure each extension whether valid or not. Please present the experiment (the methods and scenarios) with various examples and an advanced explanation.

Validity of the findings

1. In the case of Knowledge Graph (KG) let's say Wikidata. At what point does your extension improve the KG?
2. One of the critical issues in KGs is ambiguity. Can your proposed extension address this?

Additional comments

I commend the author organizes the manuscript with simple but professional language. However, the introduction and result section needs more detail in order to can improve the sound of the article.

---

## Round 0.2 · Major Revisions

Dear Author,

Please provide a valid response to the following comment from the reviewer:

There are no experiments in the article, just a few simple examples, which cannot well reflect the validity and practical significance of the research results.

Also, consider including more experimental results in your manuscript.

Thank you.

Reviewer 1 ·

Basic reporting

See additional comments.

Experimental design

See additional comments.

Validity of the findings

See additional comments.

Additional comments

The paper has been improved and I believe that it can now be accepted for publication.

---

## Round 0.3 · Major Revisions

Dear Author, Please revise and resubmit your manuscript. Also, provide a point-by-point response to the reviewers comments. Thank you.

Reviewer 3 ·

Basic reporting

It is better.

Experimental design

The experimental design is explained well

Validity of the findings

No comment

Additional comments

Thank you for the response. At the moment I just want to suggest you to a better presentation:
1. Figures used in the manuscript should be in the best resolution. Figure 5 and Figure 6 seem blur
2. The usage of caption of Immutability column on Table 1. Comparison of Knowledge Graph and Tag mechanism seems not consistent ("No" and "There is"). Why not "No" and "Yes"?

Reviewer 4 ·

Basic reporting

The work is interesting, however, I see a number of problems that, in my opinion, should be addressed. For example, the main problem I see is that the difference between Knowledge Graph Extension and Knowledge Graph Augmentation is not clear to me. The author should define both at the beginning of the paper along with examples and references, and then analyse the similarities and differences between the two to make it clear to the reader.

Experimental design

The experiment is interesting. However, it is too small to draw any conclusions. The community usually expects evaluations on a much larger scale so that there is evidence that the method can be used in the future.

Validity of the findings

Section 6 Evaluation is not really an evaluation section as such. The author briefly outlines some of the advantages of his method, but these must be supported by a large-scale quantitative evaluation (see previous point). Furthermore, it is not entirely clear how the O complexities of the worst case have been derived. And why not provide a study of the best case and the average case.

Additional comments

- The examples would be clearer if instead of using a dictionary structure they could be visualised graphically.

- The first time the expression Knowledge Graphs appears, the acronym KG can be used, and then the acronym KG can be used throughout the paper. The author has done it with other expressions, I don't understand why not in this case.

- The literature is relevant and up to date, but it is sparse and, as mentioned above, completely ignores Knowledge Graph Augmentation techniques.

Reviewer 5 ·

Basic reporting

This paper seems to present a Knowledge Graph Extensions, I'm confused about the abstract, the abstract seems to be written in some un understandable order. it is advised to rephrase the abstract in a meaningful manner to look like a abstract of a scientific paper. including some more details about experimentation and results. etc. is there any experimental done, please add in abstract too and also what's the validity of the research ?
figures are generally blur and not transparent eg. figure 4 and 5

Experimental design

The experimental details are in-sufficient to be considered it for a scientific paper. therefore, it is advised to add more experiments. there is only one experiment given with five persons e.g A,B,C,D and E. is the study only valid for this scenario or more persons and the attributes like cities, demographical factors could be considered to check the validity of your plugins ? if so, please could you add more experiments related to those scenarios ?
please also try to justify the novelty of your study, how it is better/useful than others ?

Validity of the findings

The validity of the findings are missing in general, the author needs to give a detailed evaluation mechanism and discuss in detail how the evaluations has been made and what relevant state of the art studies were discussed/used for the evaluation and justification of your study.

Additional comments

This paper seems to be revised but unfortunately i cannot see the clear revised version and track changes which the author has made. however, still the paper lacks in many aspects as discussed in above sections. Therefore, needs a major improvement.

---

## Round 0.4 · Major Revisions

Dear Author,

Please address the reviewer's comments.

Reviewer 1 ·

Basic reporting

The paper has been improved and can be accepted for publication.

Experimental design

no comment

Validity of the findings

no comment

Reviewer 4 ·

Basic reporting

The manuscript has improved in this respect.

Experimental design

The work has had problems in this regard since the first version. The experiments are not relevant enough to be considered significant by the community.

Validity of the findings

As the experiments are not sufficiently relevant, the results cannot be considered extrapolable. I do not mean that the method is not correct or appropriate, but it is not sufficiently demonstrated. Neither formally nor empirically.

Additional comments

Another problem is that the manuscript ignores all the work behind Knowledge Graph Augmentation. First of all, it is a challenge about which much has been published. But the literature provided ignores most of both classic and recent work. Secondly, it uses an unconventional nomenclature 'Knowledge Graphs Augmentations', when the community uses that expression in the singular.

---

## Round 0.5 · accepted · Accept

The author have addressed all of the reviewers' comments.

Reviewer 1 ·

Basic reporting

The author has revised the manuscript and it can now be accepted for publication.

Experimental design

no comment

Validity of the findings

no comment

Reviewer 4 ·

Basic reporting

In this new version of the manuscript, the problems that, in my opinion, were present in the previous version have been corrected.

Experimental design

In my opinion, the problems of the previous version have also been corrected in this respect.

Validity of the findings

The manuscript is, in my opinion, ready to be accepted in this respect as well.